# Overcoming the Disentanglement vs Reconstruction Trade-off via Jacobian Supervision

**José Lezama**
Universidad de la República, Uruguay
`jlezama@fing.edu.uy`

## Abstract

A major challenge in learning image representations is the disentangling of the factors of variation underlying the image formation. This is typically achieved with an autoencoder architecture where a subset of the latent variables is constrained to correspond to specific factors, and the rest of them are considered nuisance variables. This approach has an important drawback: as the dimension of the nuisance variables is increased, image reconstruction is improved, but the decoder has the flexibility to ignore the specified factors, thus losing the ability to condition the output on them. In this work, we propose to overcome this trade-off by progressively growing the dimension of the latent code, while constraining the Jacobian of the output image with respect to the disentangled variables to remain the same. As a result, the obtained models are effective at both disentangling and reconstruction. We demonstrate the applicability of this method in both unsupervised and supervised scenarios for learning disentangled representations. In a facial attribute manipulation task, we obtain high quality image generation while smoothly controlling dozens of attributes with a single model. This is an order of magnitude more disentangled factors than state-of-the-art methods, while obtaining visually similar or superior results, and avoiding adversarial training[1].

## 1 Introduction

A desired characteristic of deep generative models is the ability to output realistic images while controlling one or more of the factors of variation underlying the image formation. Moreover, when each unit in the model's internal image representation is sensitive to each of these factors, the model is said to obtain *disentangled* representations. Learning such models has been approached in the past by training autoencoders where the latent variables (or a subset of them) are constrained to correspond to given factors of variation, which can be specified (supervised) or learned from the data (unsupervised) (Bengio et al., 2013; Mathieu et al., 2016; Hu et al., 2017; Szabo et al., 2018; Kim & Mnih, 2018). The remaining latent variables are typically considered nuisance variables and are used by the autoencoder to complete the reconstruction of the image.

There exists one fundamental problem when learning disentangled representations using autoencoders, sometimes referred to as the "shortcut problem" (Hu et al., 2017; Szabo et al., 2018). If the dimension of the latent code is too large, the decoder ignores the latent variables associated to the specified factors of variation, and achieves the reconstruction by using the capacity available in the nuisance variables. On the other hand, if the dimension of the latent code is small, the decoder is encouraged to use the specified variables, but is also limited in the amount of information it can use for reconstruction, so the reconstructed image is more distorted with respect to the autoencoder's input. Szabo et al. (2018) showed that this trade-off between reconstruction and disentangling can indeed be traversed by varying the dimension of the latent code. However, no principled method exists to choose the optimal latent code dimension.

The shortcut problem was also addressed by using additional mechanisms to make sure the decoder output is a function of the specified factors in the latent code. One approach, for example, consists in swapping the specified part of the latent code between different samples, and using adversarial

---

[1]Source code available at `https://github.com/jlezama/disentangling-jacobian`.

training to make sure the output distribution is indeed conditioned to the specified factors (Mathieu et al., 2016; Lample et al., 2017; Szabó et al., 2017; Szabo et al., 2018). However, adversarial training remains a difficult and unstable optimization problem in practice.

Based on these observations, we propose a method for avoiding the shortcut problem that requires no adversarial training and achieves good disentanglement and reconstruction at the same time.

Our method consists in first training an autoencoder model, the teacher, where the dimension of the latent code is small, so that the autoencoder is able to effectively disentangle the factors of variation and condition its output on them. These factors can be specified in a supervised manner or learned from the data in an unsupervised way, as we shall demonstrate. After the teacher model is trained, we construct a student model that has a larger latent code dimension for the nuisance variables. For the student, we optimize the reconstruction loss as well as an additional loss function that constrains the variation of the output with respect to the specified latent variables to be the same as the teacher's.

In what follows, we consider autoencoder models $(E, D)$, that receive an image $\boldsymbol{x}$ as input and produce a reconstruction $\hat{\boldsymbol{x}} : D(E(\boldsymbol{x})) = \hat{\boldsymbol{x}}$. We consider that the latent code is always split into a specified factors part $\boldsymbol{y} \in \mathbb{R}^k$ and a nuisance variables part $\boldsymbol{z} \in \mathbb{R}^d$: $E(\boldsymbol{x}) = (\boldsymbol{y}, \boldsymbol{z}), D(\boldsymbol{y}, \boldsymbol{z}) = \hat{\boldsymbol{x}}$.

Consider a teacher autoencoder $(E^T, D^T)$, with nuisance variables dimension $d^T$, and a student autoencoder $(E^S, D^S)$ with nuisance variables dimension $d^S; d^S > d^T$. Because the dimension of the nuisance variables of the student is larger than in the teacher model, we expect a better reconstruction from it (i.e. $||\boldsymbol{x} - \hat{\boldsymbol{x}}^S|| < ||\boldsymbol{x} - \hat{\boldsymbol{x}}^T||$, for some norm).

At the same time, we want the student model to maintain the same disentangling ability as the teacher as well as the conditioning of the output on the specified factors. A first order approximation of this desired goal can be expressed as

$$\frac{\partial \hat{x}_j^S}{\partial y_i} \approx \frac{\partial \hat{x}_j^T}{\partial y_i}, \tag{1}$$

where $j \in \{1...H \cdot W \cdot C\}$, $H$, $W$ and $C$ are the dimensions of the output image, and $i \in \{1...k\}$ indexes over the specified factors of variation.

In this paper we propose a method to impose the first-order constraint in (1), which we term Jacobian supervision. We show two applications of this method. First, we propose an unsupervised algorithm that progressively disentangles the principal factors of variation in a dataset of images. Second, we use the Jacobian supervision to train an autoencoder model for images of faces, in which the factors of variation to be controlled are facial attributes. Our resulting model outperforms the state-of-the-art in terms of both reconstruction quality and facial attribute manipulation ability.

## 2 RELATED WORK

Autoencoders (Hinton & Salakhutdinov, 2006; Bengio et al., 2013; Kingma & Welling, 2014) are trained to reconstruct an input image while learning an internal low-dimensional representation of the input. Ideally, this representation should be disentangled, in the sense that each hidden unit in the latent code should encode one factor of variation in the formation of the input images, and should control this factor in the output images. There exist extensive literature on learning disentangled representations (Rifai et al., 2012; Bengio, 2013; Cheung et al., 2014; Kingma et al., 2014; Cogswell et al., 2015; Chen et al., 2016; Mathieu et al., 2016; Szabó et al., 2017; Perarnau et al., 2016; Hu et al., 2017; Kim & Mnih, 2018; Burgess et al., 2018).

Disentangled representations have two important applications. One is their use as rich features for downstream tasks such as classification (Rifai et al., 2012; Tran et al., 2017) or semi-supervised learning (Kingma et al., 2014). In the face recognition community, for example, disentanglement is often used to learn viewpoint- or pose-invariant features (Yang et al., 2015; Peng et al., 2017; Tran et al., 2017). A second important application is in a generative setting, where a disentangled representation can be used to control the factors of variation in the generated image (Yan et al., 2016; Perarnau et al., 2016; Higgins et al., 2016; Mathieu et al., 2016; Szabó et al., 2017; Lample et al., 2017). In this work we concentrate on the second one.

In recent years, with the advent of Generative Adversarial Networks (GANs) (Goodfellow et al., 2014), a broad family of methods uses adversarial training to learn disentangled representations

(Mathieu et al., 2016; Szabó et al., 2017; Lample et al., 2017; Perarnau et al., 2016; Chen et al., 2016). In a generative setting, the adversarial discriminator can be used to assess the quality of a reconstructed image for which the conditioning factors do not exist in the training set (Mathieu et al., 2016; Szabó et al., 2017; Chen et al., 2016).

Another alternative, proposed in Fader Networks (Lample et al., 2017), is to apply the adversarial discriminator on the latent code itself, to prevent it from containing any information pertaining to the specified factors of variation. Then, the known factors of variation or attributes are appended to the latent code. This allows to specify directly the amount of variation for each factor, generating visually pleasing attribute manipulations. Despite being trained on binary attribute labels, Fader Networks generalize remarkably well to real-valued attribute conditioning.

However, despite recent advances (Arjovsky et al., 2017; Gulrajani et al., 2017), adversarial training remains a non-trivial min-max optimization problem, that in this work we wish to avoid. Other remarkable disentangling methods that require no adversarial training are: Cheung et al. (2014), where the cross-covariance between parts of the latent representation is minimized, so that the hidden factors of variation can be learned unsupervised and Higgins et al. (2016); Kim & Mnih (2018); Burgess et al. (2018) where a factorized latent representation is learned using the Variational Autoencoder (VAE) framework. In particular, the authors of Burgess et al. (2018), propose to overcome the disentangling versus reconstruction trade-off by progressively allowing a larger divergence between the factorized prior distribution and the latent posterior in a VAE.

Related to the task of varying the factors of image generation is that of domain-transfer (Reed et al., 2015; Zhu et al., 2017; Isola et al., 2017; Choi et al., 2017; Donahue et al., 2016; Liu & Tuzel, 2016). Here the challenge is to "translate" an image into a domain for which examples of the original image are unknown and not available during training. For example, in the face generation task, the target domain can represent a change of facial attribute such as wearing eyeglasses or not, gender, age, etc. (Liu & Tuzel, 2016; Perarnau et al., 2016; Yan et al., 2016; Choi et al., 2017).

# 3 UNSUPERVISED PROGRESSIVE LEARNING OF DISENTANGLED REPRESENTATIONS

In this section we detail how the Jacobian supervision motivated in Section 1 can be applied, by ways of a practical example. We will use the Jacobian supervision to learn a disentangled image representation, where the main factors of variation are progressively discovered and learned unsupervised.

We start with a simple autoencoder model, the teacher $T$, identified by its encoder and decoder parts $(E^T, D^T)$. The output of the encoder (the latent code) is split into two parts. One part corresponds to the factors of variation $\boldsymbol{y} \in \mathbb{R}^k$ and the other part corresponds to the nuisance variables, $\boldsymbol{z} \in \mathbb{R}^d$.

We begin by using $k = 2$ and $d = 0$, meaning that the latent code of the teacher is only 2-dimensional. We consider the information encoded in these two variables as the two principal factors of variation in the dataset. This choice was done merely for visualization purposes (Figure 1).

For this example, we trained a 3-layer multi-layer perceptron (MLP) on MNIST digits, using only the L2 reconstruction loss. We used BatchNorm at the end of the encoder, so that the distribution of $\boldsymbol{y}$ is normalized inside a mini-batch. In Figure 1 (a) we show the result of sampling this two-dimensional variable and feeding the samples to the decoder $D^T$. The resulting digits are blurry, but the hidden variables learned to encode the digit class.

Next, we create a student autoencoder model $(E^S, D^S)$, similar to the teacher, but with a larger latent code. Namely, $k = 2$ and $d = 1$ instead of $d = 0$, so that the latent code has now an extra dimension and the reconstruction can be improved. In order to try to maintain the conditioning of the digit class by the 2D hidden variable $\boldsymbol{y}$, we will impose that the Jacobian of the student with respect to $\boldsymbol{y}$ be the same as that of the teacher, as in (1). How to achieve this is described next.

We take two random samples from the training set $\boldsymbol{x}_1$ and $\boldsymbol{x}_2$, and feed them to the student autoencoder, producing two sets of latent codes: $(\boldsymbol{y}_1^S, \boldsymbol{z}_1^S)$ and $(\boldsymbol{y}_2^S, \boldsymbol{z}_2^S)$, and two reconstructions $\hat{\boldsymbol{x}}_1^S$ and $\hat{\boldsymbol{x}}_2^S$, respectively. We then swap the parts of the latent code to form $(\boldsymbol{y}_2^S, \boldsymbol{z}_1^S)$ and $(\boldsymbol{y}_1^S, \boldsymbol{z}_2^S)$ and feed them to the student decoder to to obtain their respective reconstructions $\hat{\boldsymbol{x}}_{21}^S$ and $\hat{\boldsymbol{x}}_{12}^S$. We also feed

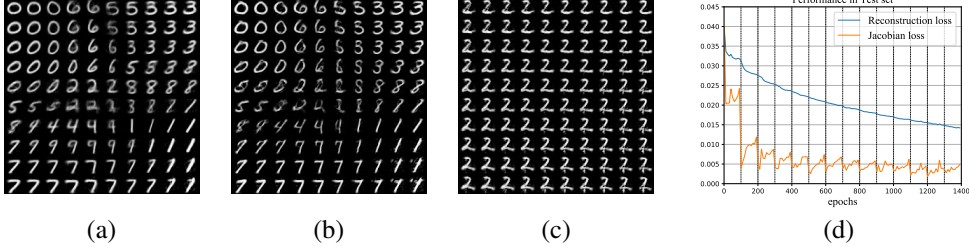

Figure 1: Unsupervised learning of disentangled representations on MNIST digits, using Jacobian supervision. **(a)** Output of teacher model ($k = 2, d = 0$) when varying its two hidden units. **(b)** Output of the final student model ($k = 2, d = 14$), while varying the same hidden units. The Jacobian supervision makes the model maintain control of the factors of variation of the teacher, while obtaining significantly better reconstruction. **(c)** A student model ($k = 2, d = 14$) trained without Jacobian supervision loses the control of the factor of variation discovered by the teacher. **(d)** Performance curves for the test set, as training of the student progresses. The gray vertical bars indicate the moments where the latent code was progressively grown by one hidden unit.

the same pair of images to the teacher autoencoder to obtain $\boldsymbol{y}_1^T, \boldsymbol{y}_2^T, \hat{\boldsymbol{x}}_1^T, \hat{\boldsymbol{x}}_2^T$. Note that the teacher encoder in this case does not produce a $\boldsymbol{z}$.

We observe, by a first-order Taylor expansion, that

$$D^T(\boldsymbol{y}_2^T) = D^T(\boldsymbol{y}_1^T) + \boldsymbol{J}_T(\boldsymbol{y}_1^T)(\boldsymbol{y}_2^T - \boldsymbol{y}_1^T) + o^T(||\boldsymbol{y}_2^T - \boldsymbol{y}_1^T||), \tag{2}$$

and

$$D^S(\boldsymbol{y}_2^S, \boldsymbol{z}_1^S) = D^S(\boldsymbol{y}_1^S, \boldsymbol{z}_1^S) + \boldsymbol{J}_S(\boldsymbol{y}_1^S, \boldsymbol{z}_1^S)\left[(\boldsymbol{y}_2^S, \boldsymbol{z}_1^S) - (\boldsymbol{y}_1^S, \boldsymbol{z}_1^S)\right] + o^S(||(\boldsymbol{y}_2^S, \boldsymbol{z}_1^S) - (\boldsymbol{y}_1^S, \boldsymbol{z}_1^S)||) \tag{3}$$

$$= D^S(\boldsymbol{y}_1^S, \boldsymbol{z}_1^S) + \boldsymbol{J}_S(\boldsymbol{y}_1^S, \boldsymbol{z}_1^S)(\boldsymbol{y}_2^S - \boldsymbol{y}_1^S, \boldsymbol{0}) + o^S(||\boldsymbol{y}_2^S - \boldsymbol{y}_1^S||), \tag{4}$$

where $\boldsymbol{J}_T$ and $\boldsymbol{J}_S$ are the Jacobian of the teacher and student decoders respectively.

Suppose

$$\boldsymbol{y}_1^S = \boldsymbol{y}_1^T \text{ and } \boldsymbol{y}_2^S = \boldsymbol{y}_2^T, \tag{5}$$

and

$$D^T(\boldsymbol{y}_2^T) - D^T(\boldsymbol{y}_1^T) = D^S(\boldsymbol{y}_2^S, \boldsymbol{z}_1^S) - D^S(\boldsymbol{y}_1^S, \boldsymbol{z}_1^S) \tag{6}$$

then, by simple arithmetic,

$$\boldsymbol{J}_S(\boldsymbol{y}_1, \boldsymbol{z}_1)\left[(\boldsymbol{y}_2 - \boldsymbol{y}_1, \boldsymbol{0})\right] \approx \boldsymbol{J}_T(\boldsymbol{y}_1)(\boldsymbol{y}_2 - \boldsymbol{y}_1), \tag{7}$$

where, since we assume (5) holds, we dropped the superscripts for clarity.

What (7) expresses is that the partial derivative of the output with respect to the latent variables $\boldsymbol{y}$ in the direction of $(\boldsymbol{y}_2 - \boldsymbol{y}_1)$ is approximately the same for the student model and the teacher model.

To achieve this, the proposed method consists essentially in enforcing the assumptions in (5) and (6) by simple reconstruction losses used during training of the student. Note that one could exhaustively explore partial derivatives in all the canonical directions of the space. In our case however, by visiting random pairs during training, we impose the constraint in (7) for random directions sampled from the data itself. This allows for more efficient training than exhaustive exploration.

Putting everything together, the loss function for training the student autoencoder with Jacobian supervision is composed of a reconstruction part $\mathcal{L}_{rec}$ and a Jacobian part $\mathcal{L}_{jac}$:

$$\mathcal{L}_{rec}(\boldsymbol{x}, E^S, D^S) := ||\boldsymbol{x} - D^S(E^S(\boldsymbol{x}))||_2^2 = ||\boldsymbol{x} - D^S(\boldsymbol{y}^S, \boldsymbol{z}^S)||_2^2 = ||\boldsymbol{x} - \hat{\boldsymbol{x}}^S||_2^2 \tag{8}$$

$$\mathcal{L}_{jac}(\boldsymbol{x}, E^S, D^S) := \lambda_y ||\boldsymbol{y}^S - \boldsymbol{y}^T||_2^2 + \lambda_{diff} || \left(D^T(\boldsymbol{y}_j^T) - D^T(\boldsymbol{y}^T)\right) - \left(D^S(\boldsymbol{y}_j^S, \boldsymbol{z}^S) - D^S(\boldsymbol{y}^S, \boldsymbol{z}^S)\right) ||_2^2 \tag{9}$$

Figure 2: $3^{rd}$ to $6^{th}$ principal factors of variation discovered by our unsupervised algorithm. The first two factors of variation are learned by the first teacher model (Figure 1 (a)). Each time a hidden unit is added to the autoencoder, a new factor of variation is discovered and learned. Each row shows the variation of the newly discovered factor for three different validation samples, while fixing all the other variables. The unsupervised discovered factors are related to stroke and handwriting style.

where the subscript $j$ indicates a paired random sample. For the experiments in Figure 1 we used $\lambda_y = 0.25, \lambda_{diff} = 0.1$. Table 4 in the appendix presents ablation studies on these hyperparameters.

In practice, we found it also helps to add a term computing the cross-covariance between $y$ and $z$, to obtain further decorrelation between disentangled features (Cheung et al., 2014):

$$\mathcal{L}_{xcov}(\boldsymbol{y}, \boldsymbol{z}) := \sum_{ij} \left[ \frac{1}{M} \sum_{m=1}^{M} \left( z_i^m - \bar{z}_i \right) \left( y_j^m - \bar{y}_j \right) \right]^2, \tag{10}$$

where $M$ is the number of samples in the data batch, $m$ is an index over samples and $i, j$ index feature dimensions, and $\bar{z}_i$ and $\bar{y}_j$ denote means over samples. In our experiments we weigh this loss with $\lambda_{xcov} = 1e^{-3}$.

Once the student model is trained, it generates a better reconstructed image than the teacher model, thanks to the expanded latent code, while maintaining the conditioning of the output that the teacher had. The extra variable in the student latent code will be exploited by the autoencoder to learn the next important factor of variation in the dataset. Examples of factors of variations progressively learned in this way are shown in Figure 2.

To progressively obtain an unsupervised disentangled representation we do the following procedure. After training of the student with $k = 2, d = 1$ is finished, we consider this model as a new teacher (equivalent to $k = 3$), and we create a new student model with one more hidden unit (equivalent to $k = 3, d = 1$). We then repeat the same procedure. Results of repeating this procedure 14 times, using 100 epochs for each stage are shown in Figure 1. In Figure 1(b), we show how the resulting final model can maintain the conditioning of the digit class, while obtaining a much better reconstruction. A model trained progressively until reaching the same latent code dimension but without Jacobian supervision, and only the cross-covariance loss for disentangling (Cheung et al., 2014), is shown in Figure 1(c). This model also obtains good reconstruction but loses the conditioning. For this model we also found $\lambda_{xcov} = 1e^{-3}$ to give the best result.

To quantitatively evaluate the disentangling performance of each model, we look at how the first two latent units ($k = 2$) control the digit class in each model. We take two images of different digits from the test set, feed them to the encoder, swap their corresponding $y$ subvector and feed the fabricated latent codes to the decoder. We then run a pre-trained MNIST classifier in the generated image to see if the class was correctly swapped. The quantitative results are shown in Table 1. We observe that the reconstruction-disentanglement trade-off is indeed more advantageous for the student with Jacobian supervision.

To complement this section, we present results of the unsupervised progressive learning of disentangled representations for the SVHN dataset (Netzer et al., 2011) in Section A.5 in the Appendix.

## 4 APPLICATION TO FACIAL ATTRIBUTE MODIFICATION

In photographs of human faces, many factors of variation affect the image formation, such as subject identity, pose, illumination, viewpoint, etc., or even more subtle ones such as gender, age, expression. Modern facial manipulation algorithms allow the user to control these factors in the generative process. Our goal here is to obtain a model that has good control of these factors and produces faithful image reconstruction at the same time. We shall do so using the Jacobian supervision introduced

Table 1: Quantitative comparison of the disentanglement and reconstruction performance of the unsupervised method on MNIST digits.

| Model | $d$ | successful class swaps | reconstruction MSE |
|---|---|---|---|
| Teacher | 0 | 94.3% | 0.036 |
| Student with Jacobian supervision | 14 | 61.7% | 0.014 |
| Student with Jacobian supervision | 18 | 52.1% | 0.012 |
| Student without Jacobian supervision | 14 | 32.6% | 0.011 |
| Random weights | 14 | 9.8% | 0.116 |

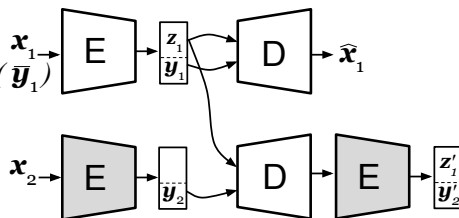

Figure 3: Diagram of the proposed training procedure for facial attributes disentangling. $E$ and $D$ always denote the same encoder and decoder module, respectively. Images $x_1$ and $x_2$ are randomly sampled and do not need to share any attribute or class. Their ground truth attribute labels are $\bar{y}_1$ and $\bar{y}_2$ respectively. The latent code is split into a vector predicting the attributes $y$ and an unspecified part $z$. Shaded $E$ indicates its weights are frozen, i.e., any loss over the indicated output does not affect its weights.

in Section 3. In this more challenging case, the disentangling will be first learned by a teacher autoencoder using available annotations and an original training procedure. After a teacher is trained to correctly disentangle and control said attributes, a student model will be trained to improve the visual quality of the reconstruction, while maintaining the attribute manipulation ability.

### 4.1 MODEL ARCHITECTURE AND LOSS FUNCTION

We begin by training a teacher model for effective disentangling at the cost of low quality reconstruction. Figure 3 shows a diagram of the training architecture for the teacher model. Let $x \in \mathbb{R}^{H \times W \times 3}$ be an image with annotated ground truth binary attributes $\bar{y} \in \{-1, 1\}^k$, where $k$ is the number of attributes for which annotations are available. Our goal is to learn the parameters of the encoder $E^T : \mathbb{R}^{H \times W \times 3} \to \mathbb{R}^{k+d}$ and the decoder $D^T : \mathbb{R}^{k+d} \to \mathbb{R}^{H \times W \times 3}$ such that $E^T(x) = (y, z)$ and $D^T(y, z) = \hat{x} \approx x$ (Figure 3, top). Ideally, $y \in \mathbb{R}^k$ should encode the specified attributes of $x$, while $z \in \mathbb{R}^d$ should encode the remaining information necessary for reconstruction.

The training of the teacher is divided into two steps. First, the autoencoder reconstructs the input $x$, while at the same time predicting in $y$ the ground truth labels for the attributes $\bar{y}$. Second, the attributes part of the latent code $y$ is swapped with that of another training sample (Figure 3, bottom). The randomly fabricated latent code is fed into the decoder to produce a new image. Typically, this combination of factors and nuisance variables is not represented in the training set, so evaluating the reconstruction is not possible. Instead, we use the same encoder to assess the new image: If the disentangling is achieved, the part of the latent code that is not related to the attributes should be the same for the existing and fabricated images, and the predicted factors should match those of the sample from which they were copied.

In what follows, we describe step by step the loss function used for training, which consists of the sum of multiple loss terms. Note that, contrary to relevant recent methods (Mathieu et al., 2016; Lample et al., 2017; Szabó et al., 2017), the proposed method does not require adversarial training.

**Reconstruction loss.** The first task of the autoencoder is to reconstruct the input image. The first term of the loss is given by the L2 reconstruction loss, as in (8).

**Prediction loss.** In order to encourage $\boldsymbol{y}$ to encode the original attributes of $\boldsymbol{x}$ indicated in the ground truth label $\bar{\boldsymbol{y}}$, we add the following penalty based on the hinge loss with margin 1:

$$\mathcal{L}_{pred}(\boldsymbol{y}, \bar{\boldsymbol{y}}) = \frac{1}{k} \sum_{i=1}^{k} \max(1 - y_i \bar{y}_i, 0), \qquad (11)$$

where the subscript $[i]$ indicates the $i^{\text{th}}$ attribute. Compared to recent related methods (Perarnau et al., 2016; Lample et al., 2017), the decoder sees the real-valued predicted attributes instead of an inserted vector of binary attribute labels. This allows the decoder to naturally learn from continuous attribute variables, leaving a degree of freedom to encode subtle variations of the attributes.

**Cycle-consistency loss.** Recall our goal is to control variations of the attributes in the generated image, with the ability to generalize to combinations of content and attributes that are not present in the training set. Suppose we have two randomly sampled images $\boldsymbol{x}_1$ and $\boldsymbol{x}_2$ as in Figure 3. After obtaining $(\boldsymbol{y}_1, \boldsymbol{z}_1) = E(\boldsymbol{x}_1)$ and $(\boldsymbol{y}_2, \boldsymbol{z}_2) = E(\boldsymbol{x}_2)$, we form the new artificial latent code $(\boldsymbol{y}_2, \boldsymbol{z}_1)$. Ideally, using this code, the decoder should produce an image with the attributes of $\boldsymbol{x}_2$ and the content of $\boldsymbol{x}_1$. Such an image typically does not exist in the training set, so using a reconstruction loss is not possible. Instead, we resort to a cycle-consistency loss (Zhu et al., 2017). We input this image to the same encoder, which will produce a new code that we denote as $(\boldsymbol{y}_2', \boldsymbol{z}_1') = E^T(D^T(\boldsymbol{y}_2, \boldsymbol{z}_1))$. If the decoder correctly generates an image with attributes $\boldsymbol{y}_2$, and the encoder is good at predicting the input image attributes, then $\boldsymbol{y}_2'$ should predict $\boldsymbol{y}_2$. We use again the hinge loss to enforce this:

$$\mathcal{L}_{cyc_1} = \frac{1}{k} \sum_{i=1}^{k} \max(1 - y_{2i} y_{2i}', 0). \qquad (12)$$

Here we could have used any random values instead of the sampled $\boldsymbol{y}_2$. However, we found that sampling predictions from the data eases the task of the decoder, as it is given combinations of attributes that it has already seen. Despite this simplification, the decoder shows remarkable generalization to unseen values of the specified attributes $\boldsymbol{y}$ during evaluation.

Finally, we add a cycle-consistency check on the unspecified part of the latent code, $\boldsymbol{z}_1$ and $\boldsymbol{z}_1'$:

$$\mathcal{L}_{cyc_2} = ||\boldsymbol{z}_1 - \boldsymbol{z}_1'||_2^2 \qquad (13)$$

**Encoder freezing.** The training approach we just described presents a major pitfall. The reversed autoencoder could learn to replicate the input code $(\boldsymbol{y}_2, \boldsymbol{z}_1)$ by encoding this information inside a latent image in whatever way it finds easier, that does not induce a natural attribute variation. To avoid this issue, a key ingredient of the procedure is to freeze the weights of the encoder when backpropagating $\mathcal{L}_{cyc_1}$ and $\mathcal{L}_{cyc_2}$. This forces the decoder to produce a naturally looking image so that the encoder correctly classifies its attributes.

**Global teacher loss.** Overall, the global loss used to train the teacher is the sum of the five terms:

$$\mathcal{L}(\theta_E, \theta_D) = \lambda_1 \mathcal{L}_{rec} + \lambda_2 \mathcal{L}_{pred} + \lambda_3 \mathcal{L}_{xcov} + \lambda_4 \mathcal{L}_{cyc_1} + \lambda_5 \mathcal{L}_{cyc_2}, \qquad (14)$$

where $\lambda_i \in \mathbb{R}, i = 1 : 5$ represent weights for each term in the sum. Details on how their values are found and how we optimize (14) in practice are described in the next section. Ablation studies showing the contribution of each loss are shown in Section A.3 in the appendix.

**Student training.** After the teacher is trained, we create a student autoencoder model with a larger dimension for the nuisance variables $\boldsymbol{z}$ and train it using only reconstruction and Jacobian supervision ((8) and (9)), as detailed in the next section.

## 4.2 IMPLEMENTATION

We implement both teacher and student autoencoders as Convolutional Neural Networks (CNN). Further architecture and implementation details are detailed in the Appendix. We train and evaluate our method on the standard CelebA dataset (Liu et al., 2015), which contains 200,000 aligned faces of celebrities with 40 annotated attributes.

The unspecified part of latent code ($\boldsymbol{z}$) of the teacher autoencoder is implemented as a feature map of 512 channels of size 2×2. To encode the attributes part $\boldsymbol{y}$, we concatenate an additional $k = 40$

channels. At the output of the encoder the values of these 40 channels are averaged, so the actual latent vector has $k = 40$ and $d = 2048$, dimensions for $\boldsymbol{y}$ and $\boldsymbol{z}$ respectively.

The decoder uses a symmetrical architecture and, following Lample et al. (2017), the attribute prediction $\boldsymbol{y}$ is concatenated as constant channels to every feature map of the decoder.

We perform grid search to find the values of the weights in (14) by training for 10 epochs and evaluating on a hold-out validation set. The values we used in the experiments in this paper are $\lambda_1 = 10^2, \lambda_2 = 10^{-1}, \lambda_3 = 10^{-1}, \lambda_4 = 10^{-4}, \lambda_5 = 10^{-5}$.

### 4.2.1 TEACHER TRAINING

At the beginning of the training of the teacher, the weights of the cycle-consistency losses $\lambda_4$ and $\lambda_5$ are set to 0, so the autoencoder is only trained for reconstruction ($\mathcal{L}_{rec}$), attribute prediction ($\mathcal{L}_{pred}$) and linear decorrelation ($\mathcal{L}_{cov}$). After 100 training epochs, we resume the training turning on $\mathcal{L}_{cyc_1}$ and $\mathcal{L}_{cyc_2}$ and training for another 100 epochs. At each iteration, we do the parameter updates in two separate steps. We first update for $\mathcal{L}_1 = \lambda_1\mathcal{L}_{rec} + \lambda_2\mathcal{L}_{pred} + \lambda_3\mathcal{L}_{cov}$. Then, freezing the encoder, we do the update (only for the decoder), for $\mathcal{L}_2 = \lambda_4\mathcal{L}_{cyc_1} + \lambda_5\mathcal{L}_{cyc_2}$.

### 4.2.2 STUDENT TRAINING

After the teacher autoencoder training is completed, we create the student model by appending new convolutional filters to the output of the encoder and the input of the decoder, so that the effective dimension of the latent code is increased.

In this experiment, we first doubled the size of the latent code from $d = 2048$ to $d = 4096$ at the $200^{th}$ epoch and then from $d = 4096$ to $d = 8192$ at the $400^{th}$ epoch. Note that this is different to the experiment of Section 3, where we grew $d$ by one unit at at time.

We initialize the weights of the student with the weights of the teacher wherever possible. Then, we train the student using the reconstruction loss (8) and the Jacobian loss (9) as defined in Section 3, using $\lambda_y = 1, \lambda_{diff} = 50$, and no prediction nor cycle-consistency loss ($\lambda_2 = \lambda_4 = \lambda_5 = 0$). The hyperparameters were found by quantitative and qualitative evaluation on a separate validation set.

### 4.3 EXPERIMENTAL RESULTS

From CelebA, we use 162,770 images of size 256x256 for training and the rest for validation. All the result figures in this paper show images from the validation set and were obtained using the same single model.

For each model, we evaluated quantitatively how well the generated image is conditioned to the specified factors. To do this, for each image in the CelebA test set, we tried to flip each of the disentangled attributes, one at a time (e.g. eyeglasses/no eyeglasses). The flipping is done by setting the latent variable $y_i$ to $-\alpha \cdot \mathrm{sign}(y_i)$, with $\alpha > 0$ a multiplier to exaggerate the attribute, found in a separate validation set for each model ($\alpha = 40$ for all models).

To verify that the attribute was successfully flipped in the generated image, we used an external classifier trained to predict each of the attributes. We used the classifier provided by the authors of Fader Networks, which was trained directly on the same training split of the CelebA dataset.

Table 2 and Figure 4 show the quantitative results we obtained. Most notably, at approximately the same reconstruction performance, the student with Jacobian supervision is significantly better at flipping attributes than the student without it. With the Jacobian supervision, the student maintains almost the same disentangling and conditioning ability as the teacher. Note that these numbers could be higher if we carefully chose a different value of $\alpha$ for each attribute.

To the best of our knowledge, Fader Networks (Lample et al., 2017) constitutes the state-of-the-art in face image generation with continuous control of the facial attributes. For comparison, we trained Fader Networks models using the authors' implementation with $d = 2048$ and $d = 8192$ to disentangle the same number of attributes as our model ($k = 40$), but the training did not converge (using the same provided optimization hyperparameters). We conjecture that the adversarial discriminator acting on the latent code harms the reconstruction and makes the optimization unstable. Comparisons with these models are shown in Table 2 and in Figures 7 and 8 in the appendix. We also show

Table 2: Quantitative comparison of the disentanglement and reconstruction performance of the evaluated models in the facial attribute manipulation task. Disentanglement is measured as the ability to flip specified attributes by varying the corresponding latent unit.

| Model | loss function | $d$ | successful attribute flips | reconstruction MSE |
|---|---|---|---|---|
| Teacher | cycle-consistency | 2048 | 73.1% | $1.82e-3$ |
| Student | Jacobian | 8192 | 72.2% | $1.08e-3$ |
| Student | cycle-consistency | 8192 | 42.7% | $1.04e-3$ |
| Fader Networks | adversarial | 2048 | 43.1% | $3.08e-3$ |
| | | 8192 | 44.2% | $1.83e-3$ |
| Random weights | | 2048 | 20.2% | $1.01e-1$ |

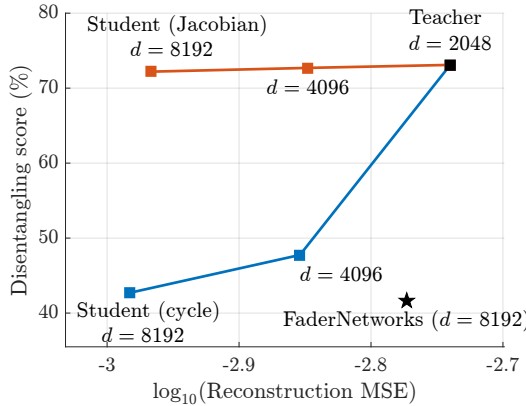

Figure 4: Disentanglement versus reconstruction trade-off for the facial attribute manipulation example (top-left is better). The disentangling score measures the ability to flip facial attributes by manipulating the corresponding latent variables.

in Figure 6 that our multiple-attribute model achieves similar performance to the single-attribute Fader Networks models provided by the authors.

Finally, Figure 5 shows the result of manipulating 32 attributes for eight different subjects, using the student model with Jacobian supervision. Note that our model is designed to learn the 40 attributes, however in practice there are 8 of them which the model does not learn to manipulate, possibly because they are poorly represented in the dataset (e.g. sideburns, wearing necktie) or too difficult to generate (e.g. wearing hat, wearing earrings).

## 5 CONCLUSION

A natural trade-off between disentanglement and reconstruction exists when learning image representations using autoencoder architectures. In this work, we showed that it is possible to overcome this trade-off by first learning a teacher model that is good at disentangling and then imposing the Jacobian of this model with respect to the disentangled variables to a student model that is good at reconstruction. The student model then becomes good at both disentangling and reconstruction. We showed two example applications of this idea. The first one was to progressively learn the principal factors of variation in a dataset, in an unsupervised manner. The second application is a generative model that is able to manipulate facial attributes in human faces. The resulting model is able to manipulate one order of magnitude more facial attributes than state-of-the-art methods, while obtaining similar or superior visual results, and requiring no adversarial training.

Figure 5: Results of attribute manipulation with the student model with Jacobian supervision. All the images were produced with the same model and belong to the test set.

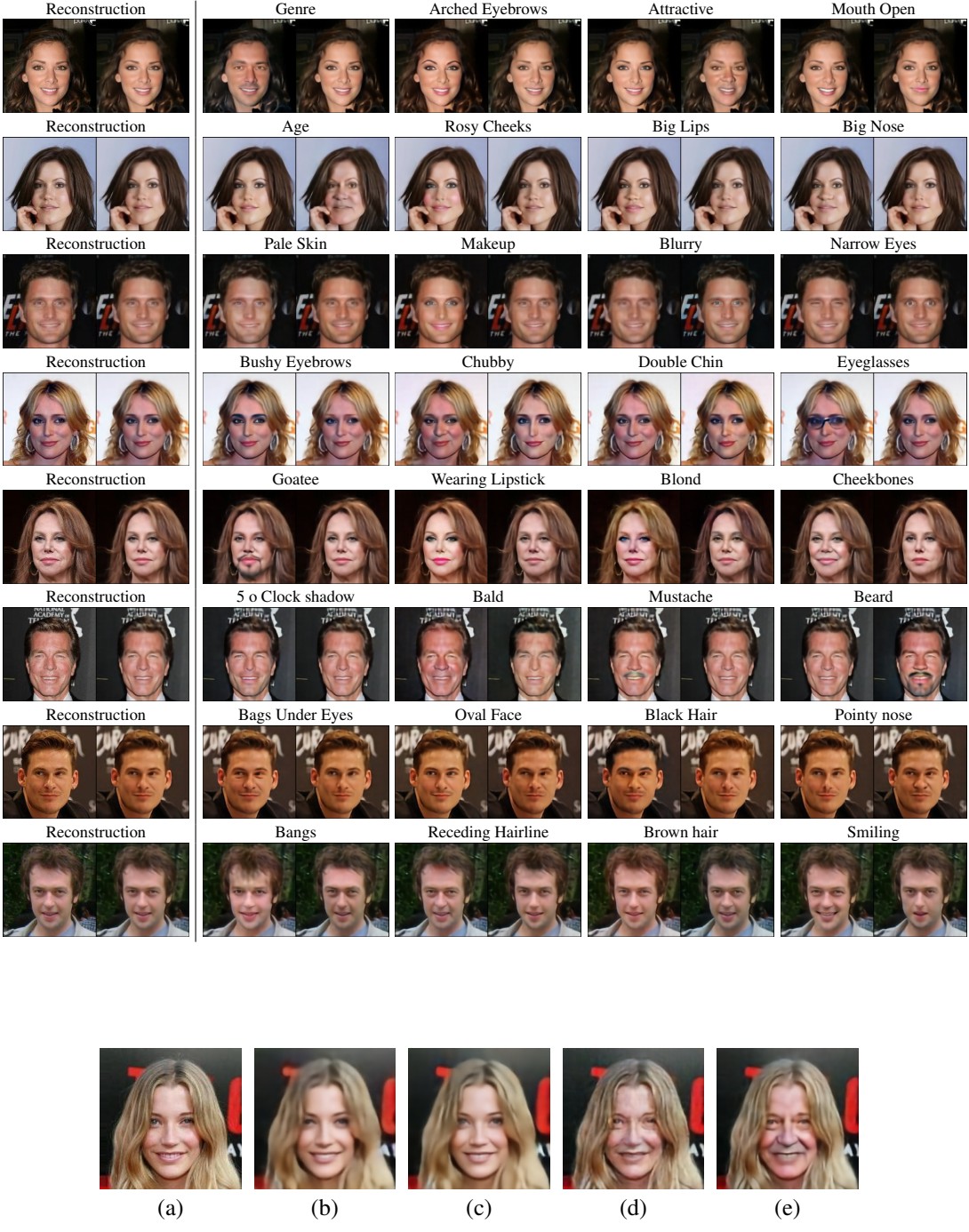

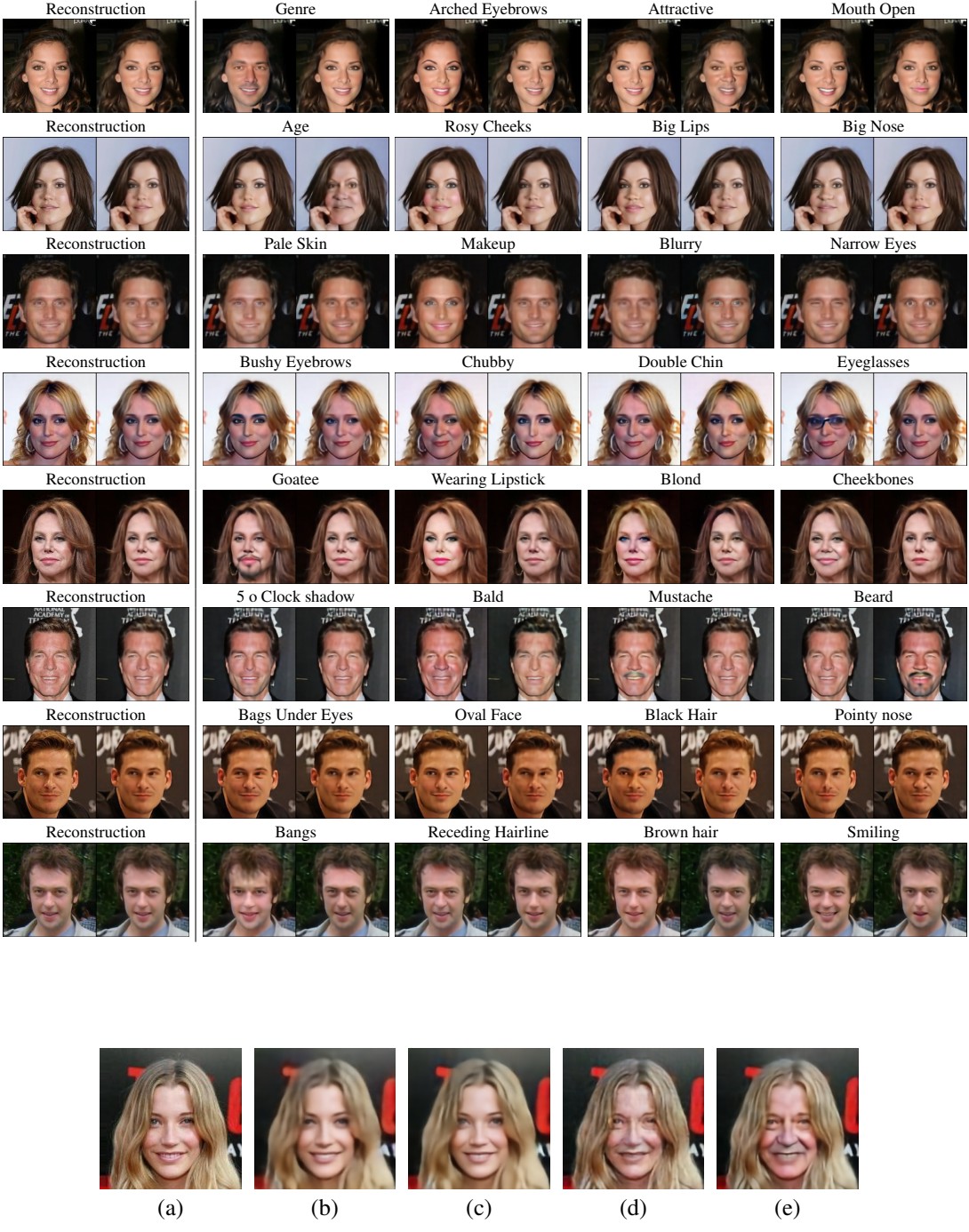

Figure 6: Comparison with single-attribute Fader Networks models (Lample et al., 2017). **(a)** Original image. **(b)** Reconstruction of Fader Networks with the provided 'eyeglasses' model. **(c)** Our teacher model achieves a sharper reconstruction using the same latent code dimension, and is able to effectively manipulate up to 32 attributes, instead of only one. **(d)** Result of amplifying age with Fader Networks with the provided aging model. **(e)** Our result for the same task.

ACKNOWLEDGMENTS

This work was supported by CAP-UDELAR Grant BPDN_2018_1. Experiments were partially run on ClusterUY, National Center for Supercomputing, Uruguay.

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

Table 3: Summary comparison of the characteristics of recent related methods. Our method has advantages over each of them, and together with Fader Networks are the only ones to generate 256×256 images while continuously varying the generated facial attributes.

| Method | end-to-end training | requires aligned pairs | requires adversarial training | face image resolution | number of attributes per model | generates continuous attributes |
|---|---|---|---|---|---|---|
| CoGAN | yes ✓ | no ✓ | yes ✗ | 128x128 ✗ | 1 ✗ | no ✗ |
| IcGAN | no ✗ | no ✓ | yes ✗ | 64x64 ✗ | 18 ✓ | no ✗ |
| Attribute2Image | no ✓ | no ✓ | no ✓ | 64x64 ✗ | 1 ✗ | yes ✓ |
| StarGAN | yes ✓ | no ✓ | yes ✗ | 128x128 ✗ | 7 ✗ | no ✗ |
| Fader Networks | yes ✓ | no ✓ | yes ✗ | 256x256 ✓ | 3 ✗ | yes ✓ |
| This work | yes ✓ | no ✓ | no ✓ | 256x256 ✓ | 32 ✓ | yes ✓ |

# A APPENDIX

## A.1 IMPLEMENTATION DETAILS FOR SECTION 3

For the autoencoder utilized for experiments in Section 3, we used the following architecture. For the encoder:

$$F(768, 256) \rightarrow ReLU \rightarrow F(256, 128) \rightarrow ReLU \rightarrow F(128, 64) \rightarrow ReLU \rightarrow FC(64, k + d)$$

where $F(I, O)$ indicates a fully connected layer with $I$ inputs and $O$ outputs. For the first teacher model ($k = 2, d = 0$), we also used BatchNorm after the encoder output.

The decoder is the exact symmetric of the encoder, with a Tanh layer appended at the end.

We used Adam (Kingma & Ba, 2014) with a learning rate of $3e^{-4}$, a batch size of 128 and weight decay coefficient $1e^{-6}$.

## A.2 IMPLEMENTATION DETAILS FOR SECTION 4

Following Lample et al. (2017), we used convolutional blocks of Convolution-BatchNorm-ReLU layers and a geometric reduction in spatial resolution by using stride 2. The convolutional kernels are all of size 4×4 with padding of 1, and we use Leaky ReLU with slope 0.2. The input to the encoder is a 256×256 image. Denoting by $k$ the number of attributes, the encoder architecture can be summarized as:

$$C(16) \rightarrow C(32) \rightarrow C(64) \rightarrow C(128) \rightarrow C(256) \rightarrow C(512) \rightarrow C(512 + k),$$

where $C(f)$ indicates a convolutional block with $f$ output channels.

The decoder architecture can be summarized as:

$$D(512 + k) \rightarrow D(512 + k) \rightarrow D(256 + k) \rightarrow D(128 + k) \rightarrow D(64 + k) \rightarrow D(32 + k) \rightarrow D(16 + k),$$

where $D(f)$ in this case indicates a deconvolutional block doing ×2 upsampling (using transposed convolutions, BatchNorm and ReLU) with $f$ input channels.

We trained all networks using Adam, with learning rate of 0.002, $\beta_1 = 0.5$ and $\beta_2 = 0.999$. We use a batch size of 128.

Table 3 shows a comparison chart between the proposed and related methods.

### A.2.1 STUDENT MODEL

For the student model, we only need to change the last layer in the encoder from $C(512 + k)$ to $C(1024 + k)$ in the first stage and $C(2048 + k)$ in the second stage. Similarly, the first layer of the decoder was changed from $D(512 + k)$ to $D(1024 + k)$ in the first stage and $D(2048 + k)$ in the second stage.

## A.3 ABLATION STUDIES

Table 4: Ablation study of the weight of each loss term for the unsupervised example of Section 3, using $k = 2$ and $d = 14$ for the student.

| $\lambda_y$ | $\lambda_{diff}$ | $\lambda_{xcov}$ | successful class swaps | reconstruction MSE |
|---|---|---|---|---|
| 0 | 0 | $1.0e^{-1}$ | 33.6% | $1.15e^{-2}$ |
| 0 | 0 | $1.0e^{-2}$ | 32.6% | $1.17e^{-2}$ |
| 0 | 0 | $1.0e^{-3}$ | 32.3% | $1.12e^{-2}$ |
| 0 | 0 | $1.0e^{-4}$ | 31.1% | $1.13e^{-2}$ |
| $2.5e^{-1}$ | $1.0e^{-1}$ | $1.0e^{-1}$ | 63.7% | $1.55e^{-2}$ |
| $2.5e^{-1}$ | $1.0e^{-1}$ | $1.0e^{-2}$ | 65.2% | $1.55e^{-2}$ |
| $2.5e^{-1}$ | $1.0e^{-1}$ | $1.0e^{-3}$ | 61.7% | $1.42e^{-2}$ |
| 1.0 | 1.0 | $1.0e^{-3}$ | 70.5% | $1.86e^{-2}$ |
| $1.0e^{-1}$ | 1.0 | $1.0e^{-3}$ | 59.4% | $1.70e^{-2}$ |
| 1.0 | $1.0e^{-1}$ | $1.0e^{-3}$ | 65.0% | $1.58e^{-2}$ |
| $1.0e^{-1}$ | $1.0e^{-1}$ | $1.0e^{-3}$ | 57.9% | $1.38e^{-2}$ |
| $1.0e^{-2}$ | $1.0e^{-1}$ | $1.0e^{-3}$ | 52.3% | $1.30e^{-2}$ |
| $1.0e^{-1}$ | $1.0e^{-2}$ | $1.0e^{-3}$ | 52.3% | $1.28e^{-2}$ |
| $1.0e^{-2}$ | $1.0e^{-2}$ | $1.0e^{-3}$ | 42.5% | $1.17e^{-2}$ |

Table 5: Ablation study of the impact of Jacobian and cycle-consistency losses in the training of the student model in the facial attribute manipulation task. The results correspond to training a student with $d = 4096$ for 50 epochs, and where the teacher was trained only with reconstruction and cross-covariance losses, with $d = 2048$. All models used the same teacher.

| $\lambda_4(\mathcal{L}_{cyc_1})$ | $\lambda_5(\mathcal{L}_{cyc_2})$ | $\lambda_{diff}$ | $\lambda_y$ | reconstruction MSE | successful flips |
|---|---|---|---|---|---|
| 0 | 0 | 50 | 1 | $1.76e - 3$ | 70.3% |
| 0 | 0 | 10 | 1 | $1.67e - 3$ | 64.8% |
| 0 | 0 | 1 | 10 | $1.85e - 3$ | 49.1% |
| 0 | 0 | 1 | 1 | $1.69e - 3$ | 46.8% |
| $1e^{-4}$ | $1e^{-5}$ | 0 | 0 | $1.78e - 3$ | 43.2% |
| $1e^{-4}$ | $1e^{-5}$ | 50 | 1 | $1.84e - 3$ | 70.7% |

Table 6: Ablation study of the weigh of the cross-covariance loss in the facial attribute manipulation example. The results correspond to training a teacher model with $d = 2048$, from scratch and for 50 epochs.

| $\lambda_{xcov}$ | reconstruction MSE | successful flips |
|---|---|---|
| $1e^{-3}$ | $2.39e - 3$ | 51.4% |
| $1e^{-2}$ | $2.50e - 3$ | 53.3% |
| $1e^{-1}$ | $2.71e - 3$ | 49.9% |

A.4   EXTENDED QUALITATIVE RESULTS FOR SECTION 4

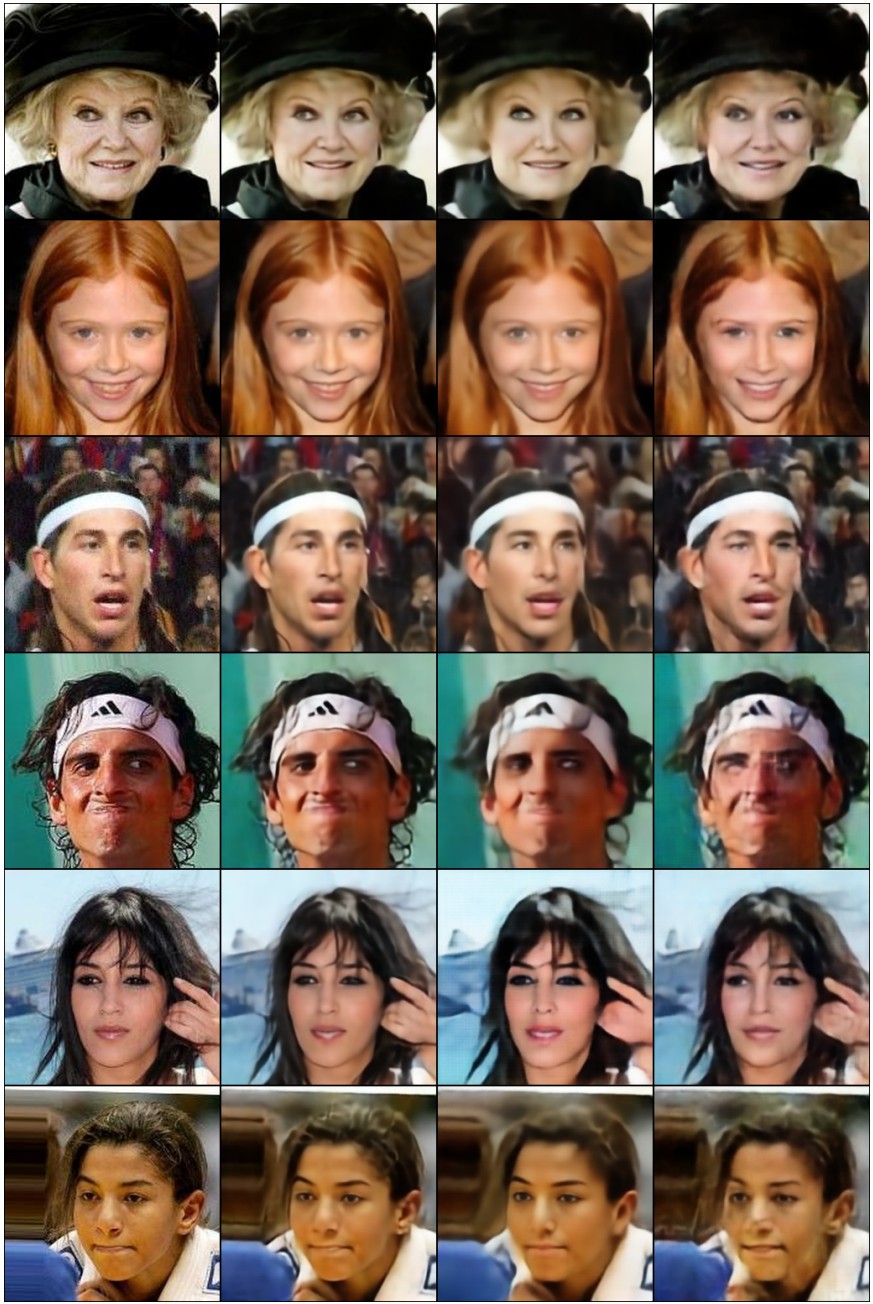

Figure 7:  Results of image reconstruction on test images. Left to right: original image, reconstruction by the student model with Jacobian supervision ($d = 8192$), by the teacher ($d = 2048$), and by the Fader Networks model trained for multiple attributes ($d = 8192$).

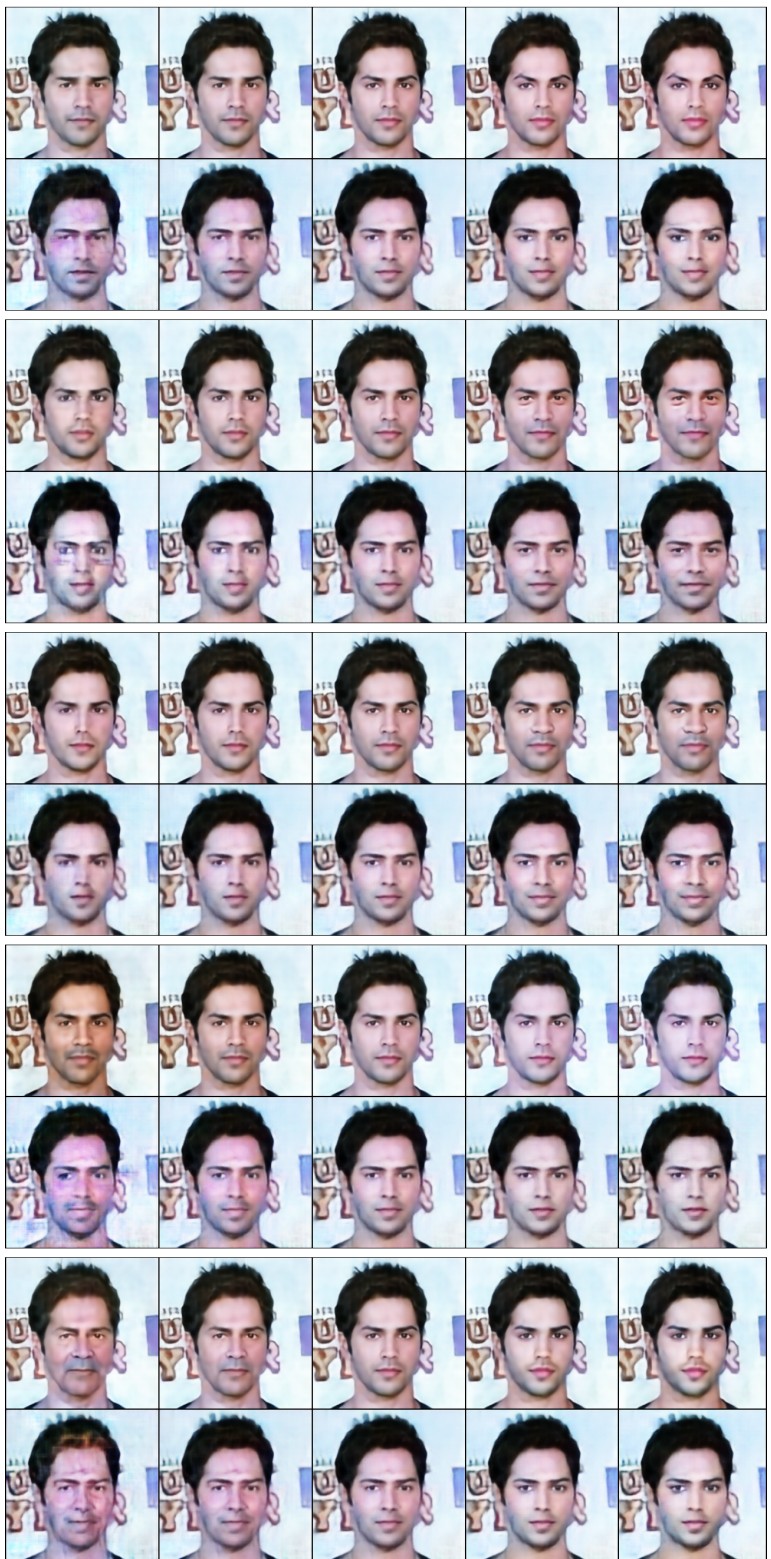

Figure 8: Results of smoothly controlling attributes for our model and Fader Networks. In each row, our result is shown on the top and Fader Networks' on the bottom. From top to bottom, the attributes are: *Arched eyebrows*, *Bags under eyes*, *Big nose*, *Pale skin*, *Age*.

### A.5 Unsupervised Progressive Learning of Disentangled Representations on SVHN Dataset

We applied the procedure described in Section 3 for progressive unsupervised learning of disentangled representations to the Street View House Numbers (SVHN) dataset (Netzer et al., 2011). The SVHN dataset contains 73,257 32×32 RGB images for training.

For this experiment, the encoder architecture is:

$$C(64) \rightarrow C(128) \rightarrow C(256) \rightarrow C(k+d) \rightarrow BatchNorm.$$

Here, $C(n)$ represents a convolutional block with $n$ $3 \times 3$ filters and zero padding, ReLU activation function and average pooling.

The decoder architecture is:

$$D(256) \rightarrow D(128) \rightarrow D(64) \rightarrow D(3).$$

Here, $D(n)$ represents a $\times 2$ upconvolution block with $n$ $4 \times 4$ filters and zero padding, ReLU activation function and average pooling.

The latent code was started with $k = 2$ and $d = 0$ and progressively grown to $k = 2, d = 16$. Each stage was trained for 25 epochs. We used $\lambda_y = 0.025, \lambda_{diff} = 0.01$. We used Adam with a learning rate of $3e - 4$, a batch size of 128 and weight decay coefficient $1e - 6$.

The first teacher model ($k = 2, d = 0$) achieves a reconstruction MSE of $1.94e^{-2}$ and the final student model ($k = 2, d = 16$) a reconstruction MSE of $4.06e^{-3}$.

Figure 9 shows the two principal factors of variation learned by the first teacher model (corresponding to $k = 2, d = 0$). Contrary to the MNIST example of Section 3, here the two main factors of variation are not related to the digit class, but to the shading of the digit. The progressive growth of the latent code is carried on from $d = 0$ to $d = 16$. The following factors of variation are related to lighting, contrast and color (see Figure 10). In this case, the unsupervised progressive method discovered factors that appear related to the digit class at the $9^{th}$ and $10^{th}$ steps of the progression. Figure 11 shows how the digit class can be controlled by the student with $d = 16$ by varying these factors. Because of the Jacobian supervision, the student is able to control the digit class while maintaining the style of the digit. Finally, in Figure 12 we show that the student also maintains control of the two main factors of variation discovered by the first teacher.



Figure 9: The two principal factors of variation learned on SVHN appear related to shading of the digit. Left to right: darker to lighter. Top to bottom: light color on the left to light color on the right.

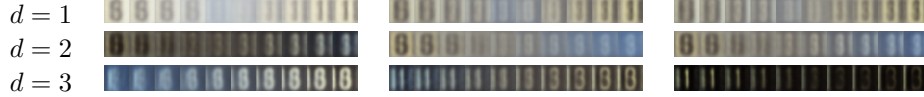

Figure 10: Third, fourth and fifth factors of variation automatically discovered on SVHN. Each row corresponds to one factor and each column corresponds to one sample. Each factor is varied while maintaining the rest of the latent units fixed.

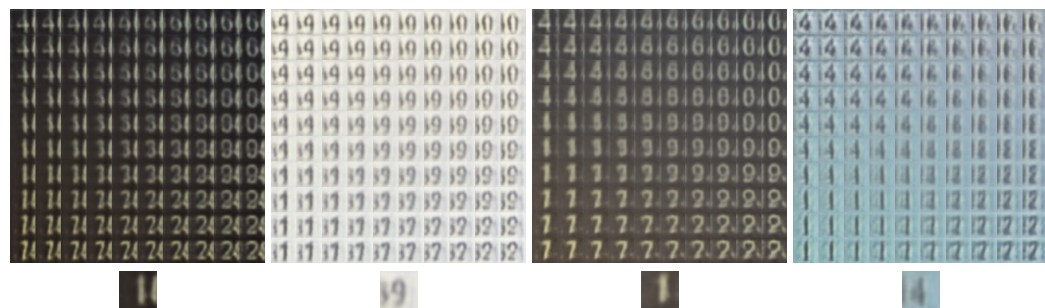

Figure 11: Factors of variation related to the center digit class appear to emerge on the 9th and 10th discovered factor during the unsupervised progressive procedure described in Section 3. Here we show how the student model with Jacobian supervision and $d = 16$ can be used to manipulate the digit class while approximately maintaining the style of the digit, by varying the latent units corresponding to those factors. The bottom row shows the original images (reconstructed by the autoencoder). All images are from the test set and were not seen during training.

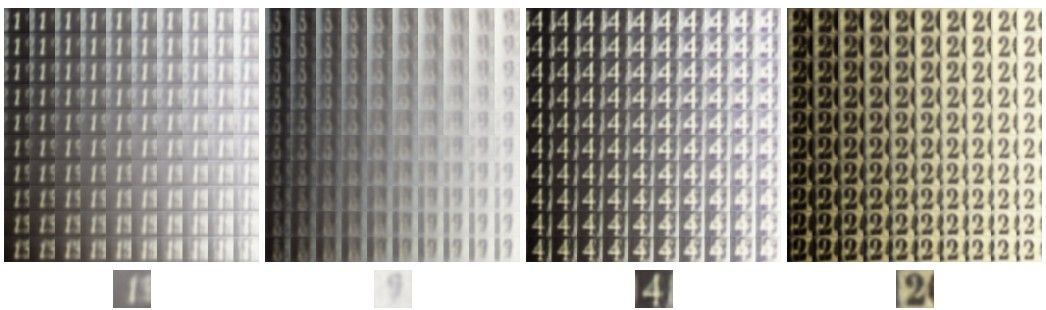

Figure 12: Result of the student with Jacobian supervision ($d = 16$) when varying the two factors learned by the teacher (Fig. 9), for four different images (whose reconstruction is shown on the bottom row). The conditioning related to shading is maintained. (Left to right: darker to lighter. Top to bottom: light color on the left to light color on the right.) All images are from the test set and were not seen during training.

