# OpenReview forum: "Overcoming the Disentanglement vs Reconstruction Trade-off via Jacobian Supervision"
_ICLR.cc/2019/Conference_

### Official Review · AnonReviewer3 · 2018-10-30
**Need more quantitative experiments to justify the claims.**

**Rating:** 5
**Confidence:** 3

**Review:**

This paper proposed a novel approach for learning disentangled representation from supervised data (x as the input image, y as different attributes), by learning an encoder E and a decoder D so that (1) D(E(x)) reconstructs the image, (2) E(D(x)) reconstruct the latent vector, in particular for the vectors that are constructed by mingling different portion of the latent vectors extracted from two training samples, (3) the Jacobian matrix matches and (4) the predicted latent vector matches with the provided attributes. In addition, the work also proposes to progressively add latent nodes to the network for training. The claim is that using this framework, one avoid GAN-style training (e.g., Fader network) which could be unstable and hard to tune.

Although the idea is interesting, the experiments are lacking. While previous works (e.g., Fader network) has both qualitative (e.g., image quality when changing attribute values) and quantitative results (e.g., classification results of generated image with novel combination of attributes), this paper only shows visual comparison (Fig. 4 and Fig. 5), and its comparison with Fader network is a bit vague (e.g., it is not clear to me why Fig. 5(e) generated by proposed approach is “more natural” than Fig. 5(d), even if I check the updated version mentioned by the authors' comments). Also in the paper there are five hyperparameters (Eqn. 14) and the center claim is that using Jacobian loss is better. However, there is no ablation study to support the claim and/or the design choice. From my opinion, the paper should show the performance of supervised training of attributes, the effects of using Jacobian loss and/or cycle loss, the inception score of generated images, etc.

I acknowledge the authors for their honesty in raising the issues of Fig. 4, and providing an updated version.

---

> ### Author Response · Authors · 2018-11-21
> **Authors' response**
>
> Thank you very much for reviewing our work.
>
>
> To address your main concern, we added quantitative comparisons by using external
> classifiers to assess the conditioning of the disentangled factors.
>
> We believe the new quantitative results strongly support our two main claims:
> 1) Our model outperforms Fader Networks by achieving better reconstruction and
>    multiple attribute manipulation.
> 2) Once a disentangling teacher model has been obtained, the proposed Jacobian
>    loss allows to add latent units that help improving the reconstruction while
>    maintaining the disentangling.
>
>
> We address each of your concerns below.
>
>
> > "e.g., it is not clear to me why Fig. 5(e) generated by proposed approach is
> “more natural” than Fig. 5(d)"
>
> We realize that this is a very subjective remark so we removed this claim from
> the image caption.  The intent of Fig. 5 is to show that even for single
> attribute manipulation and reconstruction, our proposed method performs similar
> or better than Fader Networks. For multiple attributes, a Fader Network model
> does not converge and has a poorer reconstruction and attribute manipulation
> performance. Besides the new quantitative results in Table 2 and Figure 4, this
> is also shown qualitatively in the new Figures 7 and 8 in the appendix.
>
> > "Also in the paper there are five hyperparameters (Eqn. 14) and the center
> claim is that using Jacobian loss is better. However, there is no ablation study
> to support the claim and/or the design choice."
>
> We show quantitatively in the new Table 2 and Figure 4 that using the Jacobian
> supervision performs better than the cycle-consistency loss, in terms of the
> disentanglement versus reconstruction trade-off. To measure the disentangling
> performance of the models, we manipulate the latent variables aiming to change
> the presence or absence of each attribute, and check with an external classifier
> that the attribute is indeed changed. We used a pre-trained classifier provided
> by the authors of Fader Networks.
>
> > "From my opinion, the paper should show the performance of
> supervised training of attributes, the effects of using Jacobian loss and/or
> cycle loss, the inception score of generated images, etc."
>
> We included ablation studies in the appendix (new Section A.3, page 14). These
> show the separate and combined use of Jacobian and cycle-consistency losses for
> training the student (Table 5). Their combination actually works OK. For the
> sake of simplicity we keep only the Jacobian loss, and the cycle-consistency
> loss is only used to train the disentangling by the teacher.
>
> Note that by using an external classifier, the measure we obtain is in some
> sense similar to an inception score.

---

> > ### Comment · AnonReviewer3 · 2018-11-29
> > **Better paper now.**
> >
> > The authors have done a good work to improve their submission and addressed my concerns (e.g., Tab 1 and Appendix is good). I have increased the rating by 1.

---

### Official Review · AnonReviewer1 · 2018-11-01
**Idea is neat and qualitative results are impressive, but the paper is quite lacking in quantitative results and comparisons to other methods.**

**Rating:** 7
**Confidence:** 4

**Review:**

Summary: The paper proposes a method to tackle the disentanglement-reconstruction tradeoff problem in many disentangling approaches. This is achieved by first training the teacher autoencoder (unsupervised or supervised) that learns to disentangle the factors of variation at the cost of poor reconstruction, and then distills these learned representations into a student model with extra latent dimensions, where these extra latents can be used to improve the reconstructions of the student autoencoder compared to the teacher autoencoder. The distillation of the learned representation is encouraged via a novel Jacobian loss term that encourages the change in reconstructions of the teacher and student to be similar when the latent representation changes. There is one experiment for progressive unsupervised disentangling (disentangling factor by factor) on MNIST data, and one experiment for semi-supervised disentangling on CelebA-HQ.

Pros:
- I think the idea of progressively capturing factors of variation one by one is neat, and this appears to be one of the first successful attempts at this problem.
- The distillation appears to work well on the MNIST data, and does indeed decrease the reconstruction loss of the student compared to the teacher.
- The qualitative results on CelebA-HQ look strong (especially apparent in the video), with the clear advantage over Fader Networks being that the proposed model is a single model that can manipulate the 40 different attributes, whereas Fader Nets can only deal with at most 3 attributes per model.

Cons:
- There are not enough quantitative results supporting the claim that the model is “effective at both disentangling and reconstruction.” The degree of disentanglement in the representations is only shown qualitatively via latent interpolation, and only for a single model. Such qualitative results are generally prone to cherry-picking and it is difficult to reliably compare different disentangling methods in this manner. This calls for quantitative measures of disentanglement. Had you used a dataset where you know the ground truth factors of variation (e.g. dSprites/2D Shapes data) for the unsupervised disentangling method, then the level of disentanglement in the learned representations could be quantified, and thus your method could be compared against unsupervised disentangling baselines. For the semi-supervised disentanglement example on CelebA, you could for example quantify how well the encoder predicts the different attributes (because there is ground truth here) e.g. report RMSE of the y_i’s on a held out test set with ground truth. A quantitative comparison with Fader Networks in this manner appears necessary. The qualitative comparison on a single face in Figure 5 is nowhere near sufficient.
- There is quantitative evidence that the reconstruction loss decreases when training the student, but here it’s not clear whether this quantitative difference makes a qualitative difference in the reconstructions. Getting higher fidelity images is one of the motivations behind improving reconstructions, so It would be informative to compare the reconstructions of the teacher and the student on the same image.
- In the CelebA experiments, the benefit of student training is not visible in the results. In Figure 5 you already show that the teacher model gives decent reconstructions, yet you don’t show the reconstruction for the student model (quantitatively you show that it improves in Figure 3b, but again it is worth checking if it makes a difference visually). Also it’s not clear whether Figure 4 are results from the student model or the teacher model. I’m guessing that they are from the student model.
- These quantitative results could form the basis of doing ablation studies for each of the different losses in the additive loss (for both unsupervised & semi-supervised tasks). Because there are many components in the loss, with a hyperparameter for each, it would be helpful to know what losses the results are sensitive to for the sake of tuning hyperparameters. This would be especially useful should I wish to apply the proposed method to a different dataset.
- I think the derivation of the Jacobian loss requires some more justification. The higher order terms in the Taylor expansion in (2) and (3) can only be ignored when ||y_2 - y_1|| is small compared to the coefficients, but there is no validation/justification regarding this.

Other Qs/comments:
- On page 5 in the last paragraph of section 3, you say that “After training of the student with d=1 is finished, we consider it as the new teacher”. Here do you append z to y when you form the new teacher?
- On page 6 in the paragraph for prediction loss, you say “This allows the decoder to naturally …. of the attributes”. I guess you mean this allows the model to give realistic interpolations between y=-1 and 1?
- bottom of page 6: “Here we could have used any random values in lieu of y_2” <- not sure I understand this?
- typo: conditionnning -> conditioning
- I would be inclined to boost the score up to 7 if the authors include some quantitative results along with more thorough comparisons to Fader Networks

************ Revision ***********
The authors' updates include further quantitative comparisons to Fader Networks and ablation studies for the different types of losses, addressing the concerns I had in the review. Hence I have boosted up my score to 7.

---

> ### Author Response · Authors · 2018-11-21
> **Authors' response**
>
> Thank you very much for your detailed review.
>
> We answer each item below.
>
> > "There are not enough quantitative results [...]"
>
> We added quantitative comparisons for both the unsupervised and supervised
> tasks.  The quantitative measure consists in evaluating, via an external
> classifier, how well the latent units condition the specified factor of
> variation in the generated image.
>
> In the MNIST example we measure how well the first two latent units can
> manipulate the digit class in the images generated by the student models.  The
> results are presented in the new Table 1, showing that the student with Jacobian
> supervision obtains a better trade-off between disentanglement and reconstruction.
>
> In the facial attribute manipulation task we used a pre-trained attribute
> classifier provided by the authors of Fader Networks. Using the classifier, we
> measure if by manipulating the latent unit corresponding to one attribute we can
> change the presence or absence of that attribute in the generated image. We do
> this for all attributes and for all images in the test set. The results are
> shown in Table 2 and Figure 4.
>
> For comparison, we trained two Fader Networks models to manipulate all
> attributes. The training did not converge and the resulting manipulation and
> reconstruction performance is inferior to our method. Besides the quantitative
> comparison, this can also be seen qualitatively in the new Figures 7 and 8.
>
>
> > "[...] compare the reconstructions of the teacher and the student on the same
>   image."
>
> We added a new figure to the appendix showing a comparison between the
> reconstructions obtained by the teacher and by the student (new Figure 7). It
> shows that the student model is better at reconstructing fine image details.  The
> comparison also includes a Fader Networks model trained to manipulate multiple
> attributes, and show that its reconstruction is distorted.
>
> > "Also it’s not clear whether Figure 4 are results from the student model or
>   the teacher model. [...]"
>
> Sorry for this lack of clarity. Figure 4 shows results by the student model
> trained with Jacobian supervision. We clarified this in the manuscript.
>
> > "[...]  ablation studies for each of the different losses [...]"
>
> We added ablation studies for both unsupervised and supervised tasks in the new
> section A.3 in the appendix (page 14).  Unless otherwise noted, the weighs of
> the losses were found by evaluation on separate validation sets.
>
> > "[...] The higher order terms in the Taylor expansion in (2) and (3)
>   can only be ignored when ||y_2 - y_1|| is small [...]"
>
> Indeed, because of the higher order terms, even assuming (5) and (6) hold, (7)
> is only an approximation. Note however that the norm of the approximation error
> in (7) is that of the difference between the higher order terms of the teacher
> and the student, namely ||o^T(||y_2-y_1||) - o^S(||y_2-y_1||)||. This might be
> lower than the individual higher order terms, especially if both decoders
> respond similarly to variations in $y$.  Currently, our justification is mainly
> empirical. We also considered weighing the loss by a factor reciprocal
> to ||y_2-y_1||, to give less importance to pairs of samples for
> which ||y_2-y_1|| is large.  Another option we contemplated is, for the Jacobian
> supervision, to consider a blurred version of the student, so that it has the
> low resolution of the teacher. The formulation still holds and this would also
> make (6) easier to enforce. In informal experiments we observed no significant
> advantage w.r.t. the current approach, which is simpler.  We
> leave these possible avenues of improvement as future work.
>
> > "[...] you say that “After training of
>   the student with d=1 is finished, we consider it as the new teacher”. Here do
>   you append z to y when you form the new teacher?"
>
> Yes this is correct. We clarified this in the text.
>
> > On page 6 in the paragraph for prediction loss, you say “This allows the
>   decoder to naturally …" of the attributes”. I guess you mean this allows the
>   model to give realistic interpolations between y=-1 and 1?
>
> We intended to say that we do not require the prediction to be binary values, as
> if we used the cross-entropy loss, but any real value. Thus, the decoder can
> read the amount of attribute variation from this variable, and not only if the
> attribute is present or not.
>
> > "[...] “Here we could have used any random values in lieu of y_2” [...]"
>
> We wanted to say that the $y$ part in the fabricated latent code could be
> random, but instead we sample it from the data (copy from another sample).
> We clarified this in the text.
>
> > "typo: conditionnning -> conditioning"
>
> Thank you.
>
> > "I would be inclined to boost the score up to 7 if the authors include some
>   quantitative results along with more thorough comparisons to Fader Networks"
>
> Thank you. We hope the additional quantitative and qualitative results can
> convince you of the superior performance of our method with respect to Fader
> Networks, for multiple attributes manipulation.

---

### Official Review · AnonReviewer2 · 2018-11-02
**Nice results on image manipulation**

**Rating:** 7
**Confidence:** 4

**Review:**

The paper aims to learn an autoencoder that can be used to effectively encode the known attributes/ generative factors and this allows easy and controlled manipulation of the images while producing realistic images.

To achieve this, ordinarily, the encoder produces latent code with two components y and z where y are clamped to known attributes using supervised loss while z is unconstrained and mainly useful for good reconstruction. But his setup fails when z is sufficiently large as the decoder can learn to ignore y altogether. Smaller sized z leads to poor reconstruction.

To overcome this issue, the authors propose to employ a student teacher training paradigm. The teacher is trained such that the encoder only produces y and the decoder that only consumes y. This ensures good disentanglement but poor reconstruction. Subsequently, a student autoencoder is learned which has a much larger latent code and produces both y and z. The y component is mapped to the teacher encoder’s y component using Jacobian regularization.

Positives:
The results of image manipulation using known attributes is quite impressive. The authors propose modifications to the Jacobian regularization as simple reconstruction losses for efficient training. The approach avoids adversarial training and thus is easier to train.

Negatives:
Unsupervised disentanglement results are only shown for MNIST. I am not convinced similar results for unsupervised disentanglement can be obtained on more complex datasets. Authors should include some results on this aspect or reduce the emphasis on unsupervised disentanglement. Also when studying this quantitative evaluation for disentanglement such as in beta-VAE will be nice to have.

Typos:
page 3: tobtain -> obtain
page 5: conditionning -> conditioning

---

> ### Author Response · Authors · 2018-11-21
> **Authors' response**
>
> Thank you very much for reviewing our work.
>
> We chose MNIST for the unsupervised disentangling experiment because the two
> principal factors of variation are related to the digit class and thus it served
> as a very good pedagogic example.
>
> To address your first concern, we conducted further experiments for the
> unsupervised disentanglement on the Street View House Numbers (SVHN)
> dataset. The results are shown in the appendix (Section A.5, page 17).  In this
> case, the two principal factors are related to the shading of the digit image
> and not to the class.  However, we found that later in the progressive discovery
> of factors of variation, the algorithm learns factors that are quite related to
> the digit class (ninth and tenth factors). Then, the final student model is able
> to manipulate the class of the digit while approximately maintaining the style
> of the digit (Figure 11).
>
> To address your second concern, we added quantitative experiments for the
> unsupervised example of Section 3 (new Table 1). These were obtained by using an
> external MNIST classifier to assess the digit class manipulation. The results
> show that the Jacobian supervision indeed allows a more advantageous traversing
> of the disentanglement versus reconstruction trade-off.
>
> Finally, we also added quantitative results for the CelebA experiments, showing
> the advantage of our method with respect to Fader Networks (new Table 2 and
> Figure 4).

---

### Public Comment · (anonymous) · 2018-10-02
**Interesting paper, but I have a few questions and concerns**

Dear authors, this is an interesting paper but I have a few questions and concerns:

(1) In equation (1) could you explain why y is used instead of y^S and y^T? Is y supposed to refer to some Oracle factors? And if so, it is not clear what assumption the authors are making later in the paper to relate y to y^S and y^T.

(2) In Figure 1., the authors claim that the student obtains better reconstruction than the teacher, however is there any quantitative comparison? It is not clear if Figure 1.(d) is sufficient to show this? Does epoch 0 correspond to the teacher? If it does, it would be good to say this explicitly.

(3) The derivation of Equation (7) is clear and very easy to follow.

(4) Is it possible to quantify the contribution of L_{xcov} to the model?

(5) The authors say that:
`Once the student model is trained, it generates a better reconstructed image than the teacher model, thanks to the expanded latent code, while maintaining the conditionning of the output that the teacher had.’

The authors have not quantified the level of ‘conditionning’ (disentanglement) for either the student or the teacher, so it is not clear if this claim is well backed, or the extent to which this is true. It would be hard for other researchers to build on this work, without having methods to qualitatively compare models. Higgins et al. ICLR 2017 propose one method for measuring disentanglement.

(6) A more serious concern is that the term disentanglement as defined in the abstract:

`where a subset of the latent variables is constrained to correspond to specific factors'

 is not clear nor is it consistently used throughout the paper. When the authors disentangle MNIST, they appear to be searching for linear separability, and when they disentangle CelebA they appear to be trying to assign one factor of variation (attribute) to each unit of y^T. Additionally, the paper refers more to ‘conditionning’ than disentanglement, it would be nice to rectify or explain this discontinuity between the main body of the text and the title.

(7) Reconstruction results in Figure 4. appear to be very good, however there is no quantitative evaluation nor comparison with other models.

(8) Additionally, while most of the results in Figure 4. are visually pleasing, there are no quantitative results. From these visual results it is not clear how reliably (or consistently) the model is able to edit the correct attribute?

(9) The authors say that:
`In comparison, a student model with enlarged latent code but that continues with the training procedure as the teacher, without Jacobian supervision, achieves good reconstruction but loses the effective conditionning on the attributes.’

There are no quantitative (or qualitative) results to demonstrate that the disentanglement is worse in the `student model with enlarged latent code'.

---

> ### Author Response · Authors · 2018-10-10
> **thank you for your interest**
>
> (1) In equation (1) y_i refers to an arbitrary dimension in the input space of the
> decoders. Both T and S decoders have the same input space for the specified
> variables, namely $\mathds{R}^k$.  In the paper we use the superscript when we
> want to indicate the value was produced by one of the encoders.
>
> (2) Please refer to our answer to item (5) below for a quantitative
> comparison. Yes, epoch 0 in Fig.1 (d) corresponds to the teacher. We will
> clarify it.
>
> (5) We quantified the level of disentanglement as follows: we evaluated how well
> the first two hidden variables ($k$=2), maintain the encoding of the digit class
> in the student models. We take two images of different digits from the test set,
> feed them to the encoder, swap their corresponding 2D subpart of the latent code
> and feed the fabricated latent codes to the decoder. We then run a pre-trained
> MNIST classifier in the generated image to see if the class was correctly
> swapped.
>
> | model                             | $d$ | recons. MSE | swaps OK |
> |---------------------------------+-----+---------------+-----------|
> | teacher                             |   0 |     3.66e-2 |    80.6% |
> | student w/ Jac. sup. (*) |  14 |     1.38e-2 |    57.2% |
> | student wo/ Jac. sup.     |  14 |     1.12e-2 |    32.0% |
> | student wo/ Jac. sup      |  10 |     1.40e-2 |    41.4% |
> |---------------------------------|------|--------------|------------|
> | random weights             |  14 |     1.16e-1 |     9.8% |
>
> We observe that at the same level of reconstruction performance (~1.4e-2), the
> student with Jacobian supervision maintains a better disentangling of the class
> (under this metric) than the student without it. We will include a figure
> showing that the reconstruction-disentanglement trade-off traversed by varying
> $d$ is indeed more advantageous for our model. Note that the first two variables
> do not encode perfectly the digit class. This advantage in the trade-off is much
> larger in the application of Section 4.
>
> (*) Note: this model was trained with $\lambda_{diff} = 0.1$ instead of $1.0$ as
> the one currently in the paper. The figure will be updated for this model.
>
> (4) We evaluated the disentangling measure (described in (5)), on the
> MNIST test set, for the student with Jacobian supervision:
>
> | xcov weight | $d$ | recons. MSE | swaps OK |
> |-------------+-----+-------------+----------|
> |        1e-3 |  14 |     1.38e-2 |    57.2% |
> |        1e-2 |  14 |     1.46e-2 |    56.3% |
> |        1e-1 |  14 |     1.49e-2 |    56.6% |
>
> (6) Thank you for remarking this important point.  In this paper we use the
> word disentangling to refer to both aspects:
>
> a) each latent unit in the specified part is sensitive to one generative factor
> b) the value of each of these latent units conditions the generated output such
> that it varies the corresponding generative factor
>
> We will clarify this in the manuscript and revise the text to make sure it is
> coherent.
>
> (7) See item (8)
>
> (8) We evaluated quantitatively how well the output is conditioned to the specified
> factors, similarly to the procedure described in item (5). To do this, for each
> image in the CelebA test set, we tried to flip each of the 32 disentangled
> attributes, one at a time (e.g. eyeglasses/no eyeglasses). We did the flipping
> by setting the latent variable y_i to sign(y_i)*-1*\alpha, with \alpha >0 a
> multiplier to exaggerate the attribute, found in a separate validation set for
> each model (\alpha=40 for all).
>
> To verify that the attribute was indeed flipped in the generated image, we used
> an external classifier trained to predict each of the attributes. We used the
> classifier provided by the authors of Lample et al. (2017), which was trained
> directly on the CelebA dataset.
>
> The results are as follows:
>
> | model                           |  $d$ | flips OK | recons. MSE |
> |------------------------------+--------+------------+---------------|
> | teacher                        | 2048  |    73.1% |     1.82e-3 |
> | student w/ Jac. sup.   | 8192 |    72.2% |     1.08e-3 |
> | student wo/ Jac. sup. | 8192 |    42.7% |     1.04e-3 |
> |------------------------------+--------+------------+---------------|
> | Lample et al., 2017     | 2048 |    43.1% |     3.08e-3 |
> | random weights         | 2048 |    20.2% |     1.01e-1 |
>
> At approximately the same reconstruction performance, the student with Jacobian
> supervision is significantly better at flipping attributes than the student
> without it.
>
> We also trained a Fader Networks model (Lample et al., 2017) with the same
> hyperparameters and training epochs as our teacher model. The result suggests
> that the adversarial discriminator acting on the latent code harms the
> reconstruction and that the conditionning is worse than with our teacher model.
>
> (9) We will add to the appendix the result of trying the same experiment as in
> Figure 4, but using the student model without Jacobian supervision. It will be
> clear from this experiment that the latter cannot effectively control most of
> the attributes.

---

### Author Response · Authors · 2018-11-21
**Revised version**

We thank the reviewers for their constructive comments which helped us
to significantly improve our submission.


We did the following modifications to address the reviewers concerns:

1) We addressed the lack of quantitative results, which was an
important concern shared among all reviewers. By using external classifiers on
the generated images, we were able to assess the degree of disentangling and
conditioning of the models and thus we were able to consistently quantify their
trade-off between disentanglement and reconstruction.

We believe the resulting quantitative results further support our approach.  In
particular, we quantitatively demonstrate superior performance to Fader
Networks in the facial attribute manipulation task.

2) We extended the unsupervised experiments by including results on the SVHN
   dataset (Section A.5 in the appendix, page 17).

3) We added further qualitative comparison with Fader Networks on image
   reconstrucion and attributes manipulation (Section A.4 in the appendix, page
   15).

4) We added ablation studies for the different components in the loss functions
   (Section A.3 in the appendix, page 14).

5) We replaced Figure 3(b) by a more informative graph showing the traversal of
   the disentanglement-reconstruction trade-off in the new Figure 4.



Besides the modifications suggested by the reviewers, we also did the following
changes:

6) We made minor modifications to the manuscript aiming to improve our
   exposition.

7) We use a model with different hyperparameters in Figure 1 and we corrected
   the values of two hyperparameters in the model of Section 4.

8) We added one missing reference (Burgess et al., 2018, NIPS workshops).

9) We moved  Table 3 to the appendix.

---

### Meta-Review · Area_Chair1 · 2018-12-14

**Confidence:** 4
**Recommendation:** Accept (Poster)

**Metareview:**

The paper proposes a new way to tackle the trade-off between disentanglement and reconstruction, by training a teacher autoencoder that learns to disentangle, then distilling into a student model. The distillation is encouraged with a loss term that constrains the Jacobian in an interesting way. The qualitative results with image manipulation are interesting and the general idea seems to be well-liked by the reviewers (and myself).

The main weaknesses of the paper seem to be in the evaluation. Disentanglement is not exactly easy to measure as such. But overall the various ablation studies do show that the Jacobian regularization term improves meaningfully over Fader nets. Given the quality of the results and the fact that this work moves the needle in an important (albeit hard to define) area of learning disentangled representations, I think would be a good piece of work to present at ICLR so I recommend acceptance.